# Insights on the Excitation Spectrum of Graphene Contacted with a Pt Skin

**DOI:** 10.3390/nano10040703

**Published:** 2020-04-08

**Authors:** Vito Despoja, Ivan Radović, Antonio Politano, Zoran L. Mišković

**Affiliations:** 1Institute of Physics, Bijenička 46, Zagreb HR-10000, Croatia; 2Donostia International Physics Center (DIPC), P. Manuel de Lardizabal, 20018 San Sebastian, Basque Country, Spain; 3Vinča Institute of Nuclear Sciences, University of Belgrade, P.O. Box 522, 11001 Belgrade, Serbia; iradovic@vin.bg.ac.rs; 4Department of Physical and Chemical Sciences, University of L’Aquila, via Vetoio, 67100 L’Aquila (AQ), Italy; antonio.politano@univaq.it; 5Consiglio Nazionale delle Ricerche (CNR), Istituto per la Microelettronica e Microsistemi (IMM), VIII strada 5, 95121 Catania, Italy; 6Department of Applied Mathematics, and Waterloo Institute for Nanotechnology, University of Waterloo, Waterloo, ON N2L 3G1, Canada; zmiskovi@uwaterloo.ca

**Keywords:** dirac plasmon, π plasmon, graphene, EELS, plasmonics

## Abstract

The excitation spectrum in the region of the intraband (Dirac plasmon) and interband (π plasmon) plasmons in graphene/Pt-skin terminated Pt${}_{3}$Ni(111) is reproduced by using an *ab-initio* method and an empirical model. The results of both methods are compared with experimental data. We discover that metallic screening by the Pt layer converts the square-root dispersion of the Dirac plasmon into a linear acoustic-like plasmon dispersion. In the long-wavelength limit, the Pt *d* electron excitations completely quench the π plasmon in graphene at about 4.1 eV, that is replaced by a broad peak at about 6 eV. Owing to a rather large graphene/Pt-skin separation (≈3.3 Å), the graphene/Pt-skin hybridization becomes weak at larger wave vectors, so that the π plasmon is recovered with a dispersion as in a free-standing graphene.

## 1. Introduction

Epitaxial growth on graphene represents the most suitable method to achieve large-scale samples of graphene with high crystalline quality. Moreover, graphene/metal contacts are unavoidable components in each graphene-based device. An abundant number of applications are based on collective electronic excitations, i.e., plasmonic modes. The intensity and dispersion relations of electronic excitations in supported graphene depend on the strength of the graphene interaction with the substrate [1,2,3,4,5,6,7,8]. Plasmonic excitations of graphene on different metallic surfaces, such as Ir(111), Cu(111) and Ni(111), have been studied in many experimental and theoretical reports [9,10,11,12,13,14,15,16,17,18,19,20]. The goal of most investigations on graphene/metal interfaces is to achieve a graphene sheet electronically decoupled from the substrate, with its electronic structure as similar as possible to the structure of free-standing graphene. The interest toward graphene/metal interfaces has been recently renewed by recent reports by density functional theory (DFT) indicating that a metallic surface may cause the appearance of unusual plasmonic properties in graphene modified by alkali metals [21,22,23]. Specifically, graphene doped by lithium or cesium atoms supports a very intense Dirac plasmon (DP) with its typical square-root dispersion and a weak acoustic plasmon (AP) [22,23]. When such doped graphene is placed near an Ir(111) surface, the DP is converted into acoustic-like plasmon, whereas the AP intensity is significantly enhanced [21]. From the point of view of plasmonics applications, it is very important to achieve direct interaction of two-dimensional (2D) plasmons with an external electromagnetic field. Usually this is achieved by using optically active graphene nanoparticles (nanoribbons, islands or nanorings). This enables exotic plasmonic properties such as hybridized plasmons in graphene nanostructures [24] or tunable THz electromagnetic modes in various graphene nanoparticles [25,26], which can be potentially applied in upcoming nanophotonic devices.

Another relevant open issue regarding plasmons in graphene/metal interfaces is related on the influence of screening by the metallic *d* electrons on the ultraviolet (UV) graphene plasmons, such as the interband π and π+σ plasmons. In recent years, a number of experimental and theoretical studies have been conducted in order to understand the hybridization of graphene’s π states with the metallic *d* states [27,28,29] and to elucidate collective electronic excitations in graphene/metal interfaces by means of high-resolution electron energy loss spectroscopy (EELS) [4,7,8]. This experimental technique is extremely versatile [8] and allows probing plasmon excitations from the mid/near-infrared (MIR/NIR) to the UV range of frequencies.

The EELS spectra in the MIR/NIR range are dominated by the Dirac and acoustic plasmons [1,30], which are collective electronic excitation modes related to the intraband transitions within the doped π or $\pi^{*}$ bands in the vicinity of the K-points in the Brillouin zone (BZ) of graphene. The presence of intercalated alkali-metal atoms [14,15,16,17,18,19,20], or simply the vicinity of a metal surface [27], typically give rise to significant donation of electrons into the graphene’s π band (causing up to ∼1 eV Fermi level shift relative to the Dirac point), while the graphene’s unique electronic band structure remains largely preserved. All these effects give rise to a strong DP in the graphene excitation spectrum, a clear signature of 2D character of the electron gas in graphene.

Conversely, the UV plasmonic response of graphene is dominated by a peak termed π plasmon, which results from the interband $\pi \to \pi^{*}$ transitions in the vicinity of the M-points in the BZ. This mode disperses with the in-plane momentum over a range of energies between 4 and 7 eV, depending on the type of substrate [3,4,8]. The true nature of the π plasmon resonance in free-standing graphene has been the subject of a debate over the past few years. Analyzing both the experimental spectra [7,31,32,33] and theoretical models [31,34], several issues were discussed, e.g., whether the dispersion of the π peak is linear [7] or follows a square-root dependence [32,33]. A question was also raised as to whether the π peak might even be called a plasmon mode in view of its strong character of a single-particle interband $\pi \to \pi^{*}$ transition [31,34,35]. In view of this controversy, it is interesting to further explore the nature of hybridization between the π plasmon in graphene and the electronic excitations in a metallic substrate.

In this work, we present a joint experimental and theoretical study of electronic excitations in graphene monolayer deposited on a Pt-skin-terminated Pt${}_{3}$Ni(111) surface. The main goal is to explore how the vicinity of this surface modifies the low-energy DP and the high-energy π plasmon in graphene. The former problem will be solely tackled by ab initio calculations, taking into account the effects of doping, which is responsible for the appearance of intraband electron transitions in graphene. This is achieved in our calculations by injecting extra electrons in graphene’s π band, causing shifts of the Fermi level to $E_{F}=200$ meV or $E_{F}=1$ eV relative to the Dirac point, which are typical in the presence of metallic substrate. The van der Waals nature of the graphene-Pt surface bond (with the gr-Pt separation being about h=3.3 Å) is characterized with negligible graphene/Pt-skin electronic overlap. This enables us to express the energy loss function (ELF) of the entire structure in terms of independent-electron response functions for graphene layer and the Pt-skin surface, $\chi_{i}^{0}$ with i=gr,Pt, which are obtained from independent ab initio calculations.

On the other hand, for the high-energy π plasmon, the doping effects in graphene are largely irrelevant. Besides using ab initio calculations, the ELF of graphene in the energy range near its π plasmon peak will also be described by a hydrodynamic, or Drude-Lorentz model. In that model, the graphene’s π and σ valence electrons are considered as two 2D polarizable fluids [5,36]. Such empirical treatment of the polarization of graphene is paired with an empirical model for the Pt surface response function, expressed in terms of the bulk Pt dielectric function from Refs. [37,38]. This simple empirical approach is analytically tractable and it was found to be quite effective in describing the EELS data for graphene on various metallic surfaces [39].

The rest of the paper is organized as follows. In Section 2, we present the experimental method and in Section 3 we outline our theoretical models for an effective 2D dielectric function, ϵ(Q,ω), and the corresponding ELF, $-𝕴[\epsilon^{-1}]$, for the graphene/Pt-skin-terminated Pt${}_{3}$Ni(111) interface. In Section 4, our results for the calculated ELF of the graphene/Pt-skin-terminated Pt${}_{3}$Ni(111) interface are presented and discussed, and a comparison is made with the experimental ELF spectra taken by angle-resolved EELS. Finally, we provide our concluding remarks in Section 5. Unless stated otherwise, atomic units are used, i.e., e=ℏ=m=1, where *e* is the electronic charge, ℏ is the reduced Planck constant, and *m* is the mass of electron. The graphene/Pt-skin-terminated Pt${}_{3}$Ni(111) surface will be simply called the graphene/Pt-skin for brevity.

## 2. Experimental Methods

Monolayer graphene was grown on Pt-skin terminated Pt${}_{3}$Ni(111) and characterized following procedures in Ref. [40]. Angle-resolved EELS spectra were carried out by using a Delta 0.5 spectrometer by SPECS GmbH, with energy resolution of 5 meV for experimental dataset here reported. Momentum-resolved EELS spectra were acquired by moving the analyzer while maintaining both the sample and the monochromator in a fixed position. All EELS experiments were carried out at room temperature. More details about the experimental methods can be found in Appendix A.

## 3. Theoretical Model

### 3.1. Ground State Calculation

The system we study here is shown in Figure 1. Our modeling of the EELS of this system is performed by treating graphene and the Pt-skin surface as electronically non-overlapping, i.e., chemically independent sub-systems, which only interact via the long-range Coulomb potential. Therefore, the ground-state electronic properties of both graphene and the Pt-skin are first calculated independently. In the next step, those sub-systems are coupled by the Coulomb interaction in order to determine the effective dielectric function of the entire gr/Pt-skin interface. This dielectric function is then directly used to evaluate the ELF.

The crystal structure of the Pt-skin surface is modeled by 4 atomic layers, with three atomic layers representing the Pt${}_{3}$Ni alloy cut along the (111) surface, and the fourth Pt layer being added on top of the structure, facing graphene, as shown in Figure 1. This initial (unrelaxed) surface forms a 2×2 (with respect to the Pt(111) surface) hexagonal Bravais lattice with the lattice constant $a_{S}=\sqrt{2}a_{\mathrm{Pt}}=5.49$ Å, where $a_{\mathrm{Pt}}=3.88$ Å is the lattice constant of a bulk Pt FCC crystal. The separation between atomic layers is $d=a_{\mathrm{Pt}}/\sqrt{3}=2.24$ Å. The lattice constant of graphene’s unit cell is $a_{C}=2.46$ Å. The graphene and the Pt-skin lattice constants in the direction perpendicular to the crystal planes (the lattice constants of periodic superstructures in the *z* direction) are $L_{\mathrm{gr}}=12.3$ Å and $L_{\mathrm{Pt}}=27.6$ Å, respectively.

The ground-state structural optimization and the electronic density calculations were performed using the Quantum Espresso (QE) package [41,42]. The core-electron interaction is approximated by the norm-conserving pseudopotentials [43]. The structural optimization calculation of the Pt-skin surface is performed until the maximum force on each atom was reduced below 0.002 eV/Å. The exchange–correlation (XC) potential is approximated by the Perdew-Burke-Ernzerhof generalized gradient approximation (GGA) [44]. The ground-state electronic densities of the Pt-skin and graphene are calculated using 6×6×1 and 12×12×1 Monkhorst-Pack K-point mesh [45], respectively. The separation between graphene and the topmost Pt atomic layer is fixed to h=3.3 Å [46].

### 3.2. Calculation of an Effective 2D Dielectric Function

The large equilibrium distance *h* results in negligible electronic overlap between the graphene layer and the topmost Pt layer, allowing us to perform separate calculations of the non-interacting electron response functions of graphene and the Pt-skin, thereby considerably reducing the unit cell size and saving the computational time and memory requirements. The graphene and the Pt-skin non-interacting electron response functions are
(1)$$\chi_{i}^{0}\left(Q,\omega \right)=\frac{2}{\Omega_{i}}\sum_{nm}\sum_{K\in B.Z}{\left|\rho_{nK,mK+Q}\right|}^{2}\frac{f_{nK}-f_{mK+Q}}{\omega +E_{nK}-E_{mK+Q}+i\eta \;\mathrm{sgn}\left(E_{mK+Q}-E_{nK}\right)},$$
where $\Omega_{i}=SL_{i}$ is normalization volume (with i=gr,Pt), $f_{nK}=1/\left[e^{\left(E_{nK}-E_{F}\right)/k_{B}T}+1\right]$ is the Fermi-Dirac distribution, and the charge vertices are $\rho_{nK,mK+Q}=\langle \phi_{nK}\left|e^{-iQr}\right|\phi_{mK+Q}\rangle .$ The response functions (Equation 1) are calculated using dense K-point meshes, i.e., 601×601×1 and 51×51×1 for graphene and the Pt-skin, respectively. The band summations are performed over 20 and 200 bands, for graphene and the Pt-skin, respectively. For both systems, a damping parameter of η=20 meV is used. The calculation of $\chi_{\mathrm{gr}}^{0}$ for doped graphene is performed using the rigid band approximation, such that the Fermi level is shifted to $E_{F}=200$ meV or to $E_{F}=1$ eV relative to the Dirac point.

The dynamically screened Coulomb interaction in freestanding graphene layer is
(2)$$w=v/(1-v\chi_{\mathrm{gr}}^{0,2D}),$$
where $\chi_{\mathrm{gr}}^{0,2D}=L_{\mathrm{gr}}\chi_{\mathrm{gr}}^{0}$ is graphene’s 2D response function, and v=2π/Q is bare Coulomb interaction in 2D. In the vicinity of the polarizable Pt-skin, the interaction between charge density fluctuations in graphene is mediated by the screened Coulomb interaction,
(3)$$w_{\mathrm{Pt}}=v\left(1+D_{\mathrm{Pt}}e^{-2Qh}\right)$$
which replaces the bare interaction *v* in (Equation 2). Here, the Pt-skin surface response function is defined as
(4)$$D_{\mathrm{Pt}}\left(Q,\omega \right)=\alpha \chi_{\mathrm{Pt}}^{0}/\left(1-\beta \chi_{\mathrm{Pt}}^{0}\right),$$
where the form factors $\alpha =\frac{8\pi }{Q^{3}L_{\mathrm{Pt}}}\sinh^{2}\frac{QL_{\mathrm{Pt}}}{2}$ and $\beta =\frac{4\pi }{Q^{2}}\frac{QL_{\mathrm{Pt}}+e^{-QL_{\mathrm{Pt}}}-1}{QL_{\mathrm{Pt}}}$ are corrections that result from finite thickness of the Pt-skin slab [47]. A derivation of the Pt-skin surface response function (Equation 4) is presented in Appendix B. Therefore, in the vicinity of the Pt-skin, the bare Coulomb interaction is renormalized as $v\;\;\to \;\;w_{\mathrm{Pt}}$, and the screened Coulomb interaction (Equation 2) becomes
(5)$$w=w_{\mathrm{Pt}}/\left(1-w_{\mathrm{Pt}}\chi_{\mathrm{gr}}^{0,2D}\right).$$

Finally, after using (Equation 3)–(Equation 5), the effective dielectric function for the entire gr/Pt-skin interface becomes
(6)$$\epsilon \left(Q,\omega \right)=v/w=\frac{1}{1+D_{\mathrm{Pt}}e^{-2Qh}}-v\chi_{\mathrm{gr}}^{0,2D},$$
or, in terms of the non-interacting electron response functions (Equation 1),
(7)$$\epsilon \left(Q,\omega \right)=\frac{1-\beta \chi_{\mathrm{Pt}}^{0}}{1-\beta \chi_{\mathrm{Pt}}^{0}+\alpha \chi_{\mathrm{Pt}}^{0}e^{-2Qh}}-v\chi_{\mathrm{gr}}^{0,2D}.$$

Our calculations show that, in the long-wavelength regime, Q<1/(2h), where significant hybridization takes place between graphene and the Pt-skin electronic excitations, the effects of the Pt-skin and the Pt(111) surface response functions differ only slightly. Therefore, without significant loss in accuracy, the dynamical response of the Pt-skin may be approximated by the response of a pure Pt(111) surface. Moreover, considering it as the surface of a semi-infinite Pt crystal, the surface response function may be approximated as
(8)$$D_{\mathrm{Pt}}^{\infty }=\frac{1-\epsilon_{\mathrm{Pt}}}{1+\epsilon_{\mathrm{Pt}}},$$
where $\epsilon_{\mathrm{Pt}}\left(\omega \right)$ represents the bulk Pt dielectric function in the local limit. This function may be modeled as a set of Drude-Lorentz oscillators [38],
(9)$$\epsilon_{\mathrm{Pt}}\left(\omega \right)=1-\sum_{i}\frac{f_{i}}{\omega^{2}-\omega_{i}^{2}+i\gamma_{i}\omega },$$
where $f_{i}$ represents the oscillator strength, $\omega_{i}$ is the energy of the *i*th oscillator, and $\gamma_{i}$ is the damping coefficient. Alternatively, $\epsilon_{\mathrm{Pt}}\left(\omega \right)$ could be obtained as a macroscopic dielectric function from the bulk ab initio calculations for Pt(111). After inserting (Equation 8) in (Equation 6), the effective 2D dielectric function of the entire structure may be expressed in terms of the bulk Pt dielectric function as
(10)$$\epsilon \left(Q,\omega \right)=\frac{1+\epsilon_{\mathrm{Pt}}}{2}\frac{1+\coth(Qh)}{\epsilon_{\mathrm{Pt}}+\coth\left(Qh\right)}-v\chi_{\mathrm{gr}}^{0,2D}.$$

The graphene dielectric response in the range of energies around the π plasmon peak will also be described using a two-fluid hydrodynamic model, which was very successful in modeling the EELS in graphene supported by the Pt(111), Ru(0001) and Ni(111) substrates [39]. In this model, the polarizations of graphene’s π and σ bands are described by two response functions of the Drude-Lorentz type, given by
(11)$$\chi_{\nu }\left(Q,\omega \right)=\frac{\frac{n_{\nu }^{0}Q^{2}}{m_{\nu }^{*}}}{s_{\nu }^{2}Q^{2}+\omega_{\nu r}^{2}-\omega \left(\omega +i\gamma_{\nu }\right)},$$
so that the total independent-electron 2D response function is the sum of the two component functions, $\chi_{\mathrm{gr}}^{0,2D}=\chi_{\pi }+\chi_{\sigma }$. Here, $n_{\nu }^{0}$, $m_{\nu }^{*}$, $s_{\nu }$, $\omega_{\nu r}$ and $\gamma_{\nu }$ are the equilibrium surface number density of electrons, effective electron mass, acoustic speed, restoring frequency, and the broadening parameter in the νth fluid (where ν=π,σ), respectively. In this work we use the values for these parameters that were specified in Ref. [39].

### 3.3. Electron Energy Loss Spectra

We consider a classical inelastic scattering, where the electron impinges onto the graphene surface (see Figure 1) with incident energy $E_{i}$ and is inelastically reflected from that surface with final energy $E_{f}$ without penetration of the graphene layer. Here, we assume planar scattering of the incident electron and neglect the effects of finite acceptance angle of the electron analyzer. The energy and the momentum conservation laws are,
(12)$$E_{f}=E_{i}-\omega ,$$
(13)$$Q=K_{i}\sin\theta_{i}-K_{f}\sin\theta_{f},$$
where ω is the energy loss, $K_{i,f}=\sqrt{2E_{i,f}}$ and $\theta_{i}$ and $\theta_{f}$ are the incidence and the final (scattering) angles with respect to the surface normal, as sketched in Figure 1. Using (Equation 12) and (Equation 13), the expression for the parallel momentum transfer is [1]
(14)$$Q=\sqrt{2E_{i}}\left[\sin\theta_{i}-\sqrt{1-\frac{\omega }{E_{i}}}\sin\theta_{f}\right].$$

The energy loss spectrum, which is directly probed in the reflection electron-energy-loss experiment [35], is then calculated as
(15)$$S\left(Q,\omega \right)=-\left(1/\pi \right)𝕴[\epsilon^{-1}\left(Q,\omega \right)].$$

The factor −𝕴(1/ϵ) in this expression will be called the Energy Loss Function, or the ELF.

## 4. Results and Discussion

Here, we shall first explore theoretically how the vicinity of the Pt-skin surface modifies the low-energy DP in doped graphene and the high-energy π plasmon in pristine (undoped) graphene. Then, we shall compare the results of our computations of the ELF with the spectra acquired by EELS measurements in the graphene/Pt-skin interface in the high-energy (UV) frequency range.

### 4.1. Ab Initio Results

Figure 2 shows the low-energy (IR) ELF intensities in (a) gr($E_{F}$ = 200 meV)/vacuum and (b) gr($E_{F}$ = 200 meV)/Pt-skin interfaces, the intermediate-energy (visible frequency range) ELF intensities in (c) gr($E_{F}$ = 1 eV)/vacuum and (d) gr($E_{F}$ = 1 eV)/Pt-skin interfaces and the high-energy (UV) ELF intensities in (e) gr($E_{F}$ = 0)/vacuum and (f) gr($E_{F}$ = 0)/Pt-skin interfaces. For the sake of comparison, blue circles in Figure 2b,d,f show the positions of the ELF intensity maxima from the corresponding cases of unsupported graphene in Figure 2a,c,e, respectively.

#### 4.1.1. Dirac Plasmon

From Figure 2a–d, it is obvious that, when doped graphene is deposited on the Pt-skin, the DP’s square-root dispersion (labeled DP${}_{0}$) is converted into a quasi-linear dispersion (labeled DP${}_{\mathrm{Pt}}$), i.e., the DP behaves more like an AP [48]. This behavior can be understood in simple terms by analyzing the effective dielectric function (Equation 6) in the long-wavelength limit, Q<1/(2h). The dynamical response of doped graphene can be then approximated by a Drude-type response function, $\chi_{\mathrm{Dr}}^{0}=Q^{2}\rho /\omega^{2}$, where the damping is neglected (γ=0) and ρ represents an effective concentration of charge carriers in graphene’s conduction $\pi^{*}$ band [49] around the K point in the BZ. After using this response function in (Equation 6) and retaining only the leading-order terms in *Q*, one obtains the effective dielectric function in the form
(16)$$\epsilon \approx 1/\left[1+D_{\mathrm{Pt}}\left(1-2Qh\right)\right]-2\pi Q\rho /\omega^{2}.$$

Zeros of the dielectric function in (Equation 16) determine dispersion relations of the collective electronic modes in the long-wavelength limit. Without the substrate, $\epsilon_{\mathrm{Pt}}=1$ ($D_{\mathrm{Pt}}=0$), a zero of (Equation 16) yields the well-known square-root dispersion of the DP, $\omega =\sqrt{2\pi \rho Q}$.

When graphene is in the vicinity of the Pt-skin surface, polarization of the metallic surface screens electronic excitations in graphene and thus modifies the dispersion relation of its collective modes. This modification of the DP can be understood by using a very simple qualitative picture. The DP at finite *Q* may be considered as dipolar propagating plane wave, represented by moving stripes of positive and negative charges. Because the long-wavelength DP oscillates at low (IR) frequencies, the Pt-skin surface responds instantaneously ($\omega \ll \omega_{s}$, where $\omega_{s}$ is its surface plasmon frequency), so that the surface response function $D_{\mathrm{Pt}}$ may be approximated by the image potential, or the perfect–screening model, whence $\epsilon_{\mathrm{Pt}}\to -\infty$ ($D_{\mathrm{Pt}}=-1$). This implies that the dipolar DP creates its own image in the metallic surface (exhibiting stripes of opposite charge), thereby gaining a quadrupolar character. Therefore, we expect that the proximity of the Pt surface will convert the dipolar square-root plasmon into a quadrupolar linear plasmon [50,51,52]. Indeed, when $D_{\mathrm{Pt}}=-1$ is inserted in (Equation 16), a zero of the dielectric function ϵ yields the dispersion relation $\omega =\sqrt{4\pi \rho h}\;Q$. Moreover, comparing Figure 2a,c with Figure 2b,d, it may be noticed that the Pt-skin screening also reduces the DP intensity, which is expected, given that dipolar modes generally have larger oscillatory strength.

The above simple picture also predicts that, if graphene is deposited on an insulating surface with the bulk dielectric permittivity $\epsilon_{\mathrm{ins}}>1$ ($D_{\mathrm{ins}}>-1$), a zero of (Equation 16) would still yield the square-root dispersion, $\omega \approx \sqrt{2\pi \rho }\sqrt{(1+D_{\mathrm{ins}})Q}$. Indeed, a linear DP dispersion was measured by EELS in graphene on Pt(111) [1], whereas square-root DP dispersions were observed in Refs. [53,54], where EELS measurements were performed for graphene on the SiO${}_{2}$ surface with $\epsilon_{\mathrm{ins}}\approx 4$.

#### 4.1.2. π Plasmon

Figure 2e,f demonstrate how the vicinity of the Pt-skin modifies the ELF intensity of high-energy electronic modes in pristine graphene ($E_{F}=0$). As reported in Ref. [34], in unsupported graphene and for optically small wave vectors, $Q\approx \omega_{\pi }/c$, the π plasmon is still not formed and the $\omega_{\pi }$ peak in the ELF corresponds to a large number of interband $\pi \to \pi^{*}$ electron-hole excitations around the M point of the BZ. However, for slightly larger wave-vectors ($Q>\omega_{\pi }/c$), a collective mode called π plasmon is formed, which initially exhibits linear dispersion, as experimentally observed in Refs. [7,55], and then, for larger wave vectors *Q*, it exhibits a square-root dispersion relation. This transitional behavior of the π plasmon dispersion relation is displayed by the blue dots in Figure 2f, which represent the positions of the ELF intensity maxima in the unsupported graphene (labeled $\pi_{0}$) from Figure 2e.

It can be noticed in Figure 2f that the proximity of the Pt-skin surface significantly suppresses the π plasmon spectral weight in the long-wavelength limit, Q<1/(2h). Note a dark spot in the region where the $\pi_{0}$ plasmon was located in unsupported graphene (blue circles for Q≲0.1): this effect is very similar to quenching of molecular excitons when a molecule is deposited on metallic substrate [56], suggesting that, in the long-wavelength limit, the π plasmon may indeed still be considered just an electron-hole excitation. In addition, it can be observed in Figure 2f that the spectral weight is shifted towards larger frequencies, especially in the optical limit, Q→0, whereby a broad maximum appears at $\omega_{d}\approx 6$ eV. For larger wave vectors, Q>1/(2h)∼0.15 Å${}^{-1}$, the coupling with the Pt-skin vanishes and the ELF intensity maxima in Figure 2f begin to follow the blue dots, showing that the π plasmon behaves as in the case of unsupported graphene.

The π plasmon dispersion relation can also be analyzed using the previous simplified model. The response function of graphene in the UV frequency range may be expressed by means of simplified hydrodynamic model (Equation 11), which gives in the range of frequencies close to the π plasmon $\chi_{\pi }\approx Q^{2}\rho_{\pi }/\left(\omega^{2}-\omega_{\pi }^{2}\right)$, where we have neglected the damping constant $\gamma_{\pi }=0$, and $\rho_{\pi }$ represents the corresponding effective π electron density. Using this response function in (Equation 6) and retaining the leading-order terms in *Q*, the effective dielectric function in the long-wavelength limit becomes
(17)$$\epsilon \approx 1/\left[1+D_{\mathrm{Pt}}\left(1-2Qh\right)\right]-2\pi Q\rho_{\pi }/\left(\omega^{2}-\omega_{\pi }^{2}\right).$$

Without the Pt-skin surface ($D_{\mathrm{Pt}}=0$), a zero of (Equation 17) yields the dispersion $\omega \approx \omega_{\pi }+\pi \rho_{\pi }Q/\omega_{\pi }$, which reproduces the linear behavior predicted in Ref. [34] at lower values of the wave vector *Q*. When the Pt-skin surface is present, even though at frequencies $\omega ∼\omega_{\pi }$ the Pt-surface should be treated in a dynamical regime, we may still use the perfect–screening approximation, $D_{\mathrm{Pt}}\approx -1$ (which is quite reliable near $\omega_{\pi }$, as can be seen in the lower panel in Figure 4), so that a zero of (Equation 17) yields the dispersion relation $\omega \approx \omega_{\pi }+2\pi \rho_{\pi }hQ^{2}/\omega_{\pi }$. In this case, the π plasmon dispersion relation becomes quadratic, which is indeed the trend followed by the weak ELF intensity maxima in Figure 2f.

However, it can be noticed in Figure 2f that, for Q≈0, the broad maximum (labeled $\omega_{d}$) is blue-shifted by about 2 eV with respect to the value $\omega \left(Q=0\right)=\omega_{\pi }\approx 4$ eV predicted by the above simplified model. As we shall discuss later, this is because the shifted peak is actually not related to graphene’s π plasmon, but rather corresponds to the interband $d\to d^{*}$ transitions at the Pt surface. Therefore, in the long-wavelength limit, the broad peak at frequency $\omega_{d}\approx 6$ eV in Figure 2f corresponds to the Pt $d\to d^{*}$ transitions, whereas for larger *Q*, this peak disappears and the spectra again become dominated by the $\pi_{0}$ plasmon, as in unsupported graphene. One may then conclude that the only effect of the Pt-surface is to screen and destroy the $\pi_{0}$ plasmon in the long-wavelength limit (Q≈0).

### 4.2. Comparison with Experiments for π Plasmon

Figure 3 shows the experimental EELS spectra for the graphene/Pt-skin contact. The incident electron energy is $E_{i}=70$ eV and the incidence angle is fixed at $\theta_{i}=65^{\circ }$. Eight final scattering angles have been probed: $\theta_{f}=57^{\circ }$, $59^{\circ }$, $61^{\circ }$, $63^{\circ }$, $65^{\circ }$, $67^{\circ }$, $69^{\circ }$ and $71^{\circ }$. The same experimental dataset is compared with theoretical curves for the ELF intensity, obtained by using two methods: the ab initio calculations (red-solid) and an empirical method, consisting of the two-fluid hydrodynamic model of graphene (Equation 11), paired with the surface response function of a semi-infinite Pt metal (Equation 8), which is expressed in terms of the empirical bulk Pt dielectric function (Equation 9) in the local limit (magenta-dashed). The green dashed lines represent the EELS spectra of the self-standing graphene obtained using ab initio method, for comparison.

It can be noticed that the spectra are dominated by a broad peak corresponding to the excitation of graphene’s π electron-hole transitions, or the π plasmon, which depends on the transferred wave vector *Q* and is modified by the Pt-skin surface. By analyzing the expression in (Equation 14), it may be concluded that, for the scattering angles $\theta_{f}=57^{\circ }$, $59^{\circ }$, $61^{\circ }$, $63^{\circ }$, and for energy losses ω>4 eV where the graphene π excitations appear, the momentum transfer is Q>1/(2h). Because of the $e^{-2Qh}$ factor in the effective dielectric function (Equation 6), we may expect that, for the mentioned four scattering angles, the Coulomb interaction between graphene and the Pt-skin is weak, so that the π plasmon behaves as in unsupported graphene. It can be seen that the EELS of supported (red) and self-standing (green dashed) graphene for smaller angles coincide. The nice agreement between the experimental spectra and the theoretical curves from both methods undoubtedly supports this assertion.

For scattering angles $\theta_{f}=65^{\circ }$, $67^{\circ }$, $69^{\circ }$, $71^{\circ }$, the transferred momentum approaches the optical regime, Q<1/(2h), and the interaction between graphene and the Pt-skin surface becomes stronger. One can notice fast increase of the discrepancy between the EELS of supported (red) and self-standing (green dashed) graphene as the angle increases. The strongest interaction between the π plasmon and the Pt-skin surface occurs for the two largest angles ($\theta_{f}=69^{\circ }$, $71^{\circ }$), in which case the limit Q→0 is reached in the energy-loss interval 4<ω<5 eV where the π plasmon is expected to occur, as can be seen in Figure 2e. One can notice a very good agreement with the ab initio curves for those two angles, while the empirical curves slightly underestimate the experimental π plasmon energy. This shortcoming of the empirical method is expected, because using the bulk dielectric function $\epsilon_{\mathrm{Pt}}$ in the local limit is not sufficient to describe the dispersive response of the Pt surface, as can be seen in Figure 4 below.

It should be emphasized that the present measurements, as well as other similar measurements of the π plasmon in graphene supported by metallic Pt(111) and Ni(111) surfaces [8,39], systematically find that the π plasmon energy in the optical limit, Q≈0, is more than 1 eV higher than the energy of the π plasmon in unsupported graphene, as predicted by ab initio calculations [34,57]. The reason for this discrepancy may be attributed to the random-phase approximation (RPA), used in these ab initio calculations, which does not include quasi-particle corrections of graphene’s π and $\pi^{*}$ bands and excitonic effects (interaction between excited $\pi^{*}$-electron and π-hole).

However, even high-performance ab initio calculations that include quasi-particle and excitonic G${}_{0}$W${}_{0}$-BSE corrections [58], yield the π plasmon peak at about $\omega_{p}\approx 4.6$ eV, which is still lower than the value observed in Refs. [8,39], but agrees well with other experimental measurements where the π plasmon peak was found in the energy range $4<\omega_{p}<5$ eV [7,55,59,60]. A question arises then as to why is there so large variability among different experimental measurements of the π plasmon peak position in graphene. A likely answer is that, in the experiments of Refs. [7,55,59,60], graphene was supported by insulating surfaces SiC or SiO${}_{2}$, whereas in Refs. [8,39] it was deposited on metallic surfaces Pt(111) or Ni(111). This answer is supported by the fact that strong screening, which is specific to metallic substrates, can significantly modify the π plasmon intensity in the long-wavelength limit, as may be clearly seen by comparing Figure 2e,f.

In order to further explore this argument, we show in Figure 4 the surface response function of the Pt-skin, $D_{\mathrm{Pt}}$, where we compare ab initio calculations in the long-wavelength limit (solid lines) with the empirical model based on Equations (Equation 8)–(Equation 9) (dashed lines). One notices that the ab initio Pt-skin surface excitation spectrum ($S_{\mathrm{Pt}}\propto -𝕴\left[D_{\mathrm{Pt}}\right]$) exhibits a broad peak at $\omega_{d}\approx 6$ eV, which is a consequence of the large number of interband electron transitions between the flat *d* bands in Pt [61]. On the other hand, one notices that the empirical model only exhibits a plateau in the energy range ω≳6 eV.

We argue that it is precisely those $d\to d^{*}$ transitions that screen and destroy the graphene π plasmon, thereby taking over the spectral weight and exhibiting a broad peak at $\omega_{d}\approx 6$ eV.

This can be clearly seen by comparing Figure 2f and the upper panel in Figure 4, which reveals that the non-dispersive $\omega_{d}$-peak in Figure 2f is actually the $\omega_{d}$ peak from ab initio calculations in Figure 4. Accordingly, if the peaks in Figure 3e–h were generated by the π plasmon in graphene, they would be more dispersive. In other words, the observed non-dispersive nature of those peaks points to their likely origin being in the non-dispersive *d* electron transitions related to the Pt-skin support.

A feasible experimental scenario may be therefore rationalized in two steps. First, in the long-wavelength limit ($\theta_{f}\geq 65^{\circ }$), the *d* electron-hole transitions at the Pt surface totally screen the long-wavelength π plasmon, so that the *d* transitions take over the entire spectral weight. Therefore, the experimental peaks in Figure 3g,h very likely correspond to the excitations of the Pt *d* electrons, giving rise to the $\omega_{d}$ peak in Figure 4. Second, for larger wave vectors *Q* ($\theta_{f}\leq 63^{\circ }$), the interaction of graphene with the Pt-skin is weak, so that the experimental peaks in Figure 3a–d correspond to the excitation of the π plasmon, which behaves as in unsupported graphene.

## 5. Conclusions

In this paper, we provided a joint theoretical/experimental investigation of the excitation spectrum of the graphene/Pt-skin interface. We assessed the role played by the Pt-skin surface in modifying the low-energy (IR) Dirac plasmon in doped graphene, and the high-energy (UV) π plasmon in pristine graphene. It was demonstrated that the metallic surface converts the square-root Dirac plasmon dispersion into a linearly dispersing plasmon, as previously observed experimentally. It was further shown that, in the optical limit, Q<1/(2h), the Pt $d\to d^{*}$ electron-hole excitations destroy the π plasmon in graphene and take the overall spectral weight. For larger wave vectors, Q>1/(2h), the interaction with the Pt-skin is weak, so that the π plasmon is restored with features closely resembling those in freestanding graphene.

## Figures and Tables

**Figure 1 nanomaterials-10-00703-f001:**
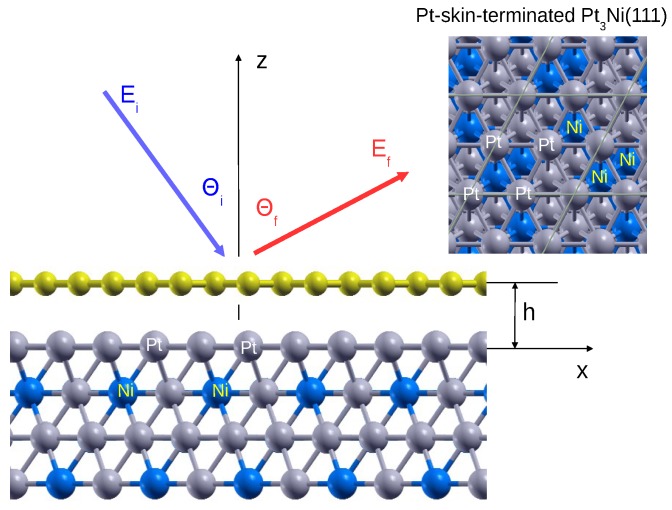
Crystal structure of graphene deposited on the Pt-skin terminated Pt${}_{3}$Ni(111).

**Figure 2 nanomaterials-10-00703-f002:**
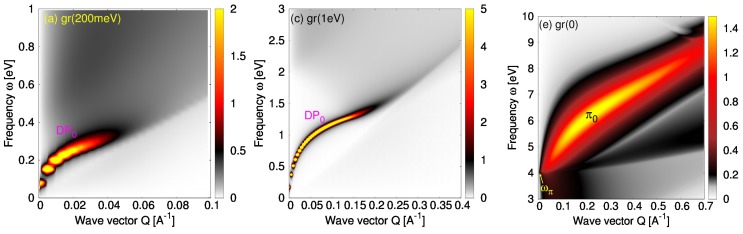
The low-energy (IR) energy loss function (ELF) intensities, −𝕴(1/ϵ), in (**a**) gr($E_{F}=$ 200 meV)/vacuum and (**b**) gr($E_{F}=$ 200 meV)/Pt-skin interfaces, the intermediate-energy (VIS) ELF intensities in (**c**) gr($E_{F}=1$ eV)/vacuum and (**d**) gr($E_{F}=1$ eV)/Pt-skin interfaces and high-energy (UV) ELF intensities in (**e**) gr($E_{F}=0$)/vacuum and (**f**) gr($E_{F}=0$)/Pt-skin interfaces. The blue circles in the panels (**b**,**d**,**f**) show the positions of the ELF intensities maxima in the unsupported graphene cases displayed in the panels (**a**,**c**,**e**), respectively. The graphene Fermi energy $E_{F}$ is given relative to the Dirac point.

**Figure 3 nanomaterials-10-00703-f003:**
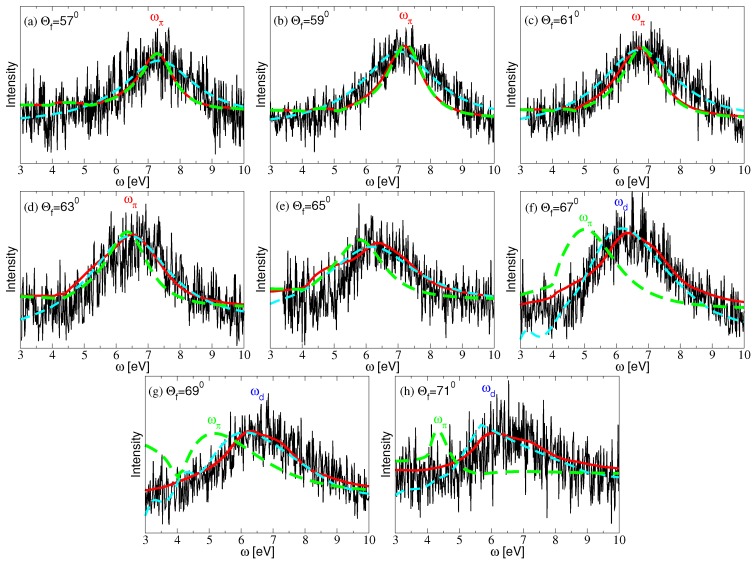
The experimental EELS spectra of the graphene/Pt-skin interface for various final (scattering) angles: (**a**) $\theta_{f}=57^{\circ }$–(**h**) $\theta_{f}=71^{\circ }$ (in steps of $2^{\circ }$). The incidence angle is $\theta_{i}=65^{\circ }$ and the incident electron energy is $E_{i}=70$ eV. The experimental data are compared with theoretical results for the energy loss function, −𝕴(1/ϵ), obtained using two methods: ab initio calculations (red-solid) and an empirical model (magenta-dashed).The green dashed lines represent the EELS spectra of the self-standing graphene obtained using ab initio method, for comparison.

**Figure 4 nanomaterials-10-00703-f004:**
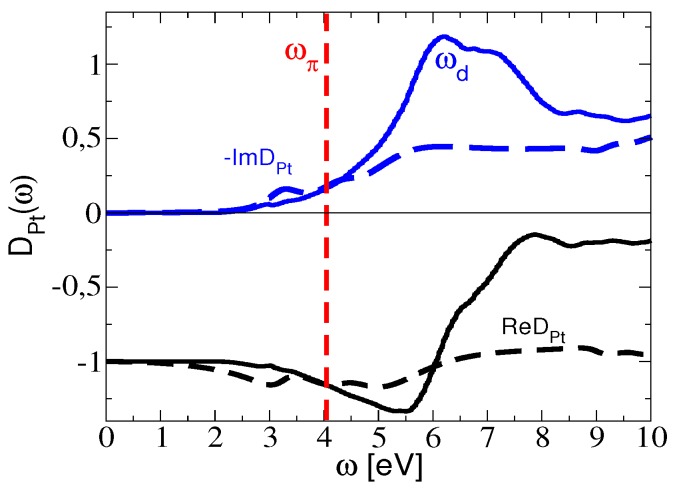
The ab initio (solid) and the empirical (dashed) Pt-skin surface response function, $D_{\mathrm{Pt}}\left(\omega \right)$, for Q≈0. The vertical dashed line shows the energy of the π plasmon in unsupported graphene.

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
