# Peer review of "Insights on the Excitation Spectrum of Graphene Contacted with a Pt Skin"

_nanomaterials, 2020, doi:10.3390/nano10040703_

Round 1
Reviewer 1 Report
This manuscript deals with the combine experimental and theoretical investigation of the excitation spectrum of the graphene/Pt-skin interface. The authors presented the angle-resolved electron energy loss spectroscopy spectra and the experimental observations were mimic both with using ab-initio simulations and an empirical model.
The authors demonstrated that the metallic surface converts the square-root Dirac plasmon dispersion into a linearly dispersing plasmon. This was previously observed experimentally. It was also found, that in the optical limit, electron-hole excitations destroy the π plasmon in graphene and take the overall spectral weight. For larger wave vectors, the interaction with the Pt-skin is weak, so that the π plasmon is restored with features closely resembling those in freestanding graphene.
This manuscript contains interesting new theoretical works. The topic of the manuscript is fit to the scope of “Nanomaterials” and I recommend it for publication, however, before publication the authors can take into consideration the followings:
- Although the manuscript focus on the theoretical descriptions, it stated that the work contains combined experimental and theoretical works. However the experimental part is very short. Please extend, and describe the experimental method significantly started with the sample preparation, testing of the sample and related topics.
- Please indicate why the authors used this holder for the single layer graphene compared with the well know and used other substrates.
- Is there a possibility to extend the energy range of the measured spectra at least till 100 eV? In other words, can the authors able to measure the spectra with higher electron incident energy. The simulations of the electron energy spectra at low incident energies are more challenges usually than at higher energies. Also to go further to obtain the energy loss function of the 2D materials required a larger energy loss range.
- Can the authors give estimation of the bulk contribution of the substrate for the present incident energy range?
- The authors concluded that “for larger wave vectors, the interaction with the Pt-skin is weak, so that the π plasmon is restored with features closely resembling those in freestanding graphene.” Can the authors give direct comparison with the free standing graphene?
Reviewer 2 Report
In this work, "Insights on the excitation spectrum of graphene contacted with a Pt skin", the authors presented the excitation spectrum of graphene/Pt-skin in the region of the intraband and interband plasmons both using ab-initio method and an empirical model. Based on the obtained data (by comparing with the experimental results), the authors claimed that the metallic screening by the Pt layer converts the square-root dispersion of the intraband plasmonics into a linear acoustic-like plasmon dispersion. This manuscript has a strong potential for a second review after applying the issues and addressing the shortcomings listed below:
1-The authors should polish/revise some grammatical mistakes and typos along the manuscript. For instance, ‘…have been recently extensively studied…’.
2-In the Introduction section, while describing some of the important points in graphene/metal interfaces, potential application areas of graphene should also mentioned. To this end, the following works should be considered and cited, to give a more general view to the possible readers of the work: [(i) Opt. Mater. 73, 729 (2017); (ii) ACS Appl. Electron. Mater. 1, 637-641 (2019); (iii) Nanoscale 11, 8091-8095 (2019)].
3-A simple schematic can be added to the Experimental Methods section, to visualize what have been done experimentally (or on the characterization side).
4-In Figure 2, make the numbers on the color bars bigger.
5-In Figures 3e and 3f, it seems ωd is missing.
Round 2
Reviewer 2 Report
In its current form, the revised manuscript is suitable for publication.